# Dysfunctional Breathing, in COPD: A Validation Study

**DOI:** 10.3390/jcm14072353

**Published:** 2025-03-29

**Authors:** Andreas Daskalakis, Irini Patsaki, Aikaterini Haniotou, Emmanouil Skordilis, Afrodite Evangelodimou, Eirini Grammatopoulou

**Affiliations:** 1Laboratory of Advanced Physiotherapy, Department of Physiotherapy, University of West Attica, 122 43 Egaleo, Greece; adaskalakis@uniwa.gr (A.D.); aevangelodimou@uniwa.gr (A.E.); igrammat@uniwa.gr (E.G.); 2Department of Respiratory Medicine, General Oncologic Hospital ‘’St. Anargyroi’’, 145 64 Kifisia, Greece; haniotoy@hol.gr; 3School of Physical Education and Sport Sciences, National and Kapodistrian University of Athens, 172 37 Dafni, Greece; eskord@phed.uoa.gr

**Keywords:** Nijmegen Questionnaire, validity, reliability, COPD

## Abstract

The present study examined the validity and reliability of the Nijmegen Questionnaire, a tool used for screening dysfunctional breathing (DB) in people with chronic obstructive pulmonary disease (COPD). DB for COPD patients was described with respiratory symptoms and symptoms caused by central and peripheral tetany. The NQ value “>23” was found to be the cutoff score to detect the presence of DB in COPD. DB for the specific sample was detected in over half of the participants. Moreover, the higher NQ score was associated with a higher level of COPD severity, frequency of acute exacerbations, and increased risk of hospitalization and death. In addition, the higher the NQ score, the greater the disease burden, including symptoms such as coughing, phlegm, chest tightness, breathlessness while walking or climbing stairs, and reduced energy and confidence in leaving home. These findings confirm that the questionnaire can be a useful screening tool for COPD patients, helping healthcare providers identify and manage breathing issues more effectively. This could lead to better treatment and improved quality of life for people with COPD.

## 1. Introduction

Dysfunctional breathing (DB) has no formal definition to date. DB is suggested as “an alteration in the normal breathing patterns that results in dyspnea or other respiratory and/or non-respiratory chronic symptoms” [1]. Boulding et al. [2] indicate a variety of DB types such as hyperventilation syndrome (HVS), thoracic dominant breathing, sporadic deep sighing, forced abdominal expiration, and thoracoabdominal asynchrony. The mechanism of breathing dysfunction remains unclear [2]. Thus, healthcare professionals fail to provide appropriate treatment.

In chronic obstructive pulmonary disease (COPD), pulmonary hyperinflation, respiratory muscle weakness, which reduces inspiratory time, and the sensation of airway obstruction might be key factors associated with dyspnea [3]. Dyspnea is the main symptom of COPD that involuntarily leads to alternative breathing patterns [4,5], such as thoracic dominant breathing with forced abdominal expiration [2]. Finally, in COPD, dyspnea and DB appear to trigger each other [6].

Currently, there is no gold-standard tool for diagnosing DB. Several methods have been proposed for the detection of hyperventilation syndrome (HVS) such as the cardio-pulmonary exercise test [7], the hyperventilation provocation test [8,9], optoelectronic plethysmography, the Nijmegen Questionnaire (NQ) [10], as well as the Hi–Lo test [11] and the Manual Assessment of Respiratory Motion (MARM) [12] for the evaluation of the biomechanical dimension of DB. The NQ has been used worldwide for over 40 years as a screening tool for DB and has been validated primarily in people with HVS [10,13], and secondary in people with asthma [14]. The prevalence of DB in patients with asthma, measured with the NQ, has been found to be 34% [14], while in COPD, it was nearly 50%, measured with the nonvalidated NQ [4]. The NQ comprises 16 items under three factors: (a) shortness of breath, (b) peripheral tetany and (c) central tetany [13].

Based on the above, the aim of the present study was to validate the NQ in people with stable COPD and to detect the prevalence of DB for the specific population. Thus, healthcare professionals involved in the treatment of COPD will have access to a valid and reliable tool both in the assessment of these patients and in their holistic rehabilitation.

## 2. Material and Methods

### 2.1. Study Population

The present study was conducted from January 2022 to July 2023. A sample of 84 outpatients (minimum 5 individuals per item) [15] of the Respiratory Medicine Clinic, General Oncology Hospital of Kifisia, Athens, Greece, all with diagnosed stable COPD (not experiencing acute exacerbation), participated in the study [16]. Specifically, 55 (65.5%) were men and 29 (34.5%) women, aged from 41 to 80 years (Mean = 67.68, SD = 7.97). In total, 33 (39.3%) of them were cachectic, 35 (41.7%) had normal weight, 13 (15.5%) were overweight, and 3 (3.6%) were obese. As for COPD severity, 6 participants (7.1%) suffered from mild COPD, 34 (40.5%) had moderate COPD, 32 (38.1%) had severe COPD, and 12 (14.3%) exhibited very severe COPD. Moreover, 79 (94%) participants were nonsmokers and only 5 (6%) were smokers [17]. Exclusion criteria were asthma, physical and mental disability, cardiovascular diseases, as well as musculoskeletal and neurological disorders. All patients signed an informed consent for their enrolment in the study. The study protocol was approved by the Research Ethics Committee of the (a) General Oncology Hospital of Kifissia, Athens, Greece (No 1261/17-11-2020), and (b) University of West Attica (No 29509/31-03-2021).

### 2.2. Research Tools

The Nijmegen Questionnaire (NQ);The BODE index;The COPD Assessment Test (CAT);The Borg dyspnea scale.-*The NQ* is a 16-item screening tool for DB [10], responded with a 5-point Likert-type format ranging from 0 (never) to 4 (very often). The NQ describes respiratory, cardiovascular, neurological, gastrointestinal, and psychological symptoms [13]. It has been validated in a sample of healthy people with HVS and revealed three factors labeled (a) shortness of breath, (b) peripheral tetany, and (c) central tetany [13]. It has also been validated in people with stable asthma and revealed a single-factor model with 11 items and a cutoff score of “>23” [14]. No validation study has been conducted for people with COPD so far.-*The body mass index, airflow obstruction, dyspnea, and exercise (BODE index)* predicts COPD severity, number of acute exacerbations and mortality with better accuracy than the forced expiratory volume in 1 s (FEV1) alone [18]. The BODE index is a multidimensional index that combines four independent predictors: body mass index (BMI) (kg/m^2^) [17], degree of airflow obstruction (percentage of FEV_1_), breathlessness as measured by the modified Medical Research Council dyspnea scale (MRC) [19,20], and exercise ability as determined by the 6 min walk distance test (6MWDT) [21]. The BODE index has provided validity and reliability evidence in people with COPD [22]. Celli et al. [18] described that a score of “>5” indicates a higher risk of mortality, and Global Initiative for Chronic Obstructive Lung Disease (GOLD) [16] proposed that a cutoff score of “>7” highlights severe prognosis (1-year mortality ~30–40%); this cutoff is often used as a criterion for advanced therapies like lung transplantation [16,18].-*The COPD Assessment Test (CAT)* was used to evaluate the impact of COPD on the patient’s health status and its change across time [23]. The instrument consists of eight items. The total CAT score is the sum score of the 8 items (scored 0–5) and ranges from 0 to 40. CAT has provided evidence of validity and reliability in people with COPD. CAT scores of 10–20 indicate a medium impact of COPD on health status, while scores of 20–30 and “>30” indicate high and very high impact, respectively [24].-*The Borg dyspnea scale* was used to define the perceived degree of dyspnea of the participants. Its score ranges from 0: minimum score to 10: maximum score [25], with the higher score indicating the worse intensity of dyspnea. The Borg scale has shown reproducibility in people with COPD [26].-*Dysfunctional breathing* was diagnosed by a pneumonologist who initially examined the presence of apical breathing, and/or forced expiration with abdominal muscle contraction at rest [2] and then evaluated the presence of at least 5 criteria of a 10-criterion list for the referred patients at rest [27]. Specifically, the 10-criterion list includes (a) sense of inspiratory heaviness; (b) sense of not being able to take deep breaths; (c) increased breathing frequency (>16/min); (d) frequent sighing/yawning; (e) frequent need to clear the throat; (f) muscle and joint tenderness in the upper part of the chest (sternocostal joints and intercostal muscles); (g) hacking cough; (h) chest tightness; (j) sensation of lump in the throat; and (k) previous or current effects of stress. The pneumonologist was blinded to the participant’s NQ score.

The pneumonologist diagnosed dysfunctional breathing, measured lung function (FEV_1_), and confirmed the severity of COPD [16]. The main researcher administered all the questionnaires in a random order and collected the data. Data were analyzed by a professional.

## 3. Statistical Analysis

Data processing was performed with the IBM SPSS program, version 29. The validation of the NQ was examined by construct, convergent, divergent and discriminant validity as well as internal consistency.

*The construct validity* of the NQ, up to now, has been reported for adults from (a) the “general population with HVS”, with 16 items under three factors [10] and (b) the “asthma” population, with 11 items under a single factor [14]. To identify the factorial structure of the NQ for the specific sample with stable COPD, we decided to proceed to a principal axis factoring analysis with an oblimin rotation. Preliminary tests included: (a) the Bartlett test of sphericity and (b) the KMO (Kaiser–Meyer–Olkin) test of sampling adequacy [28]. The number of factors was estimated with the following criteria: (a) the eigenvalues >1.00, (b) the scree plot, (c) the percentage % of explained variability, and (d) the content of each factor. The perspective items per factor were kept according to criteria such as (a) factor loading “>0.40”, and item communality h^2^ > 0.40 [28]. Differences between known groups provided further evidence for the construct validity, using *t*-tests and one-way ANOVAs [29].

*The divergent validity* was tested through correlations of the NQ total score with the score of the BODE index, Borg dyspnea scale and CAT (Pearson’s *r* correlation coefficient) [29].*The convergent validity* was examined through the correlation of the ΝQ total score with the pneumonologist’s diagnosis (Pearson’s r correlation coefficient) [29].*The discriminant validity testing* was conducted through the Receiver Operating Characteristic (ROC) analysis, using the pneumonologist’s diagnosis as the criterion [2,27].*The reliability testing of the NQ* was calculated through the internal consistency (Cronbach’s alpha reliability coefficient) [29].

## 4. Results

### 4.1. Participants

The 84 participants of the present study displayed the following mean scores: (a) NQ 27.25 (±12.66), (b) FEV1% predicted 50.45 (±18.89), (c) Borg scale 2.34 (±1.68), (d) CAT 24.38 (±10.96), (e) 6MWDT 313 (±100.53), (f) MRC 2.38 (±1.19), (g) BMI 25.71 (±4.93), (h) BODE index 3.87 (±2.41).

The 49 participants with DB presented the following mean scores: (a) NQ 36.67 (±6.65), (b) FEV1% predicted 37.51 (±9.21), (c) Borg scale 2.94 (±1.52), (d) CAT 25.77 (±5.70), (e) 6MWDT 271.06 (±88.60), (f) MRC 2.92 (±0.90), (g) BMI 25.32 (±4.40), (h) BODE index 5.35 (±1.74). DB was detected (NQ score “>23”) (a) in 16 women and 33 men, and (b) in 12 patients with very severe COPD, in 30 patients with severe COPD, and in 7 patients with moderate COPD.

### 4.2. Validation

#### 4.2.1. Construct Validity

##### Principal Axis Factoring Analysis

The KMO criterion (0.916) confirmed the sample adequacy for the analysis. The Bartlett index of sphericality was statistically significant (1360, *df* 120, *p* < 0.001). Section Principal Axis Factoring Analysis revealed a three-factor model, with eigenvalues above 1, explaining 74.7% of the variability. All 16 question items of the original NQ were retained, as they met the specific predefined criteria. Cronbach’s alpha was found to be 0.97, 0.88 and 0.75 for the first, second and third factors, respectively, while for the NQ, the total score was 0.94. Factor loadings are presented in Table 1.

##### Differences Between Known Groups

Significant differences were found between COPD people with and without DB (*p* < 0.001) as well as among participants of all COPD levels of severity (GOLD, 2023) (*F* = 84.47, *p* < 0.001). Post hoc analysis with Bonferroni adjustment (0.05/6 = 0.008) showed significant differences between (a) mild and moderate (*p* < 0.001), severe (*p* < 0.001) and very severe COPD (*p* < 0.001), (b) moderate and severe (*p* < 0.001) and very severe COPD (*p* < 0.001) and (c) severe and very severe COPD (*p* < 0.001). In general, the participants with mild COPD scored significantly lower than the participants of the other levels of severity. The participants with very severe COPD had the highest score among all severity levels. No evidence of significant differences between (a) smokers and non-smokers with COPD (*p* = 0.05), (b) men and women with COPD (*p* = 0.31), or (c) people with COPD who have follow-up every 1–3 months/year and in deterioration of the symptoms (*p* = 0.05) was noted. The results are presented in Table 2.

#### 4.2.2. Convergent Validity Testing

A high correlation was found between the NQ score with the pneumonologist’s rating (*r* = 0.98).

#### 4.2.3. Divergent Validity Testing

The positive correlations of the total NQ score with BODE index (*r* = 0.81), CAT (*r* = 0.49) and Borg dyspnea scale (*r* = 0.47) showed that the higher the total NQ score, the higher the BODE index, the CAT score, and the perceived dyspnea (Borg). The results of divergent validity testing are presented in Table 3.

#### 4.2.4. Discriminant Validity Testing

The ROC curve analysis indicated the NQ value of >23 as the optimal cutoff score with 95.92 sensitivity and 94.29 specificity. The area under the ROC curve (AUC) was found 0.98 (*p* < 0.0001), significantly different from AUC = 0.5 (Appendix A in Appendix A), presenting the ability of the NQ to discriminate COPD people who exhibit DB from those with no DB. The results of discriminant analysis are presented in Table 4.

##### Prevalence of Dysfunctional Breathing

The prevalence of DB for the specific sample with stable COPD, according to the cutoff score > 23 for the 16-item NQ, was found to be 58.3%.

## 5. Discussion

This study presented the first validity and reliability evidence for the NQ in individuals with stable COPD. A three-factor model was identified: Factor 1 with nine items, factor 2 with four items and factor 3 with three items.

The first factor expresses “shortness of breath” in COPD and refers to respiratory symptoms, such as chest pain (No1); feeling tense (No2); accelerated or deepened breathing (No6); shortness of breath (No7); constricted chest (No8); bloated abdominal sensation (No9); unable to breathe deeply (No11); palpitation (No15); and feeling of anxiety (No16). Specifically, chest pain is a common symptom, due to gastroesophageal reflux disease, fibrotic pleura, vagal hyperactivity, bronchial spasm, diaphragm dysfunction, phrenic neuropathy, and fascial strain, with prevalence between 22% and 55% in people with COPD [30]. Feelings of tension and anxiety are also present in COPD patients who experience distress and impaired function [31]. The items of faster or deeper breathing, shortness of breath, and unable to breathe deeply address the dominant upper thoracic pattern (rapid and shallow breathing) in COPD patients [32]. A bloated feeling in the stomach is the sensation of a hyperinflated chest due to pulmonary hyperinflation, which causes dyspnea [33,34]. The constricted chest might be explained by abnormal alterations in intercostal muscles and the changes in chest biomechanics in people with COPD [35]. Finally, the sense of palpitations is a common hypoxemia-related COPD symptom [36]. The structure of the first factor of the NQ for COPD patients revealed some differences compared to that for the general population with HVS [13]. Specifically, item No9 (bloated abdominal sensation) for COPD patients was classified under the first factor (respiratory symptoms), while in the study of VHS was under the third factor (central tetany). Further, item No16 (feeling of anxiety), in our study, was grouped under the first factor, while in the study of HVS, it was not under any factor. Since bloated abdominal sensation is a manifestation of breathlessness in COPD, and the feeling of anxiety reflects the sensation of respiratory limitations, these two items may represent respiratory symptoms in people with COPD.

The second factor states “peripheral tetany” in COPD with tingling fingers (No10); stiff fingers or arms (No12); tight feelings around the mouth (No13); and cold hands or feet (No14). Specifically, all these symptoms indicate peripheral tetany in COPD that follows a different pathophysiological pathway, which is driven by systemic inflammation and hypoxia [37], compared to hyperventilation in HVS [13] and asthma [14]. However, the structure and content of the second factor in the present study align with those described by van Dixhoorn and Duivenvoorden [13].

The third factor reflects “central tetany” in COPD with blurred vision (No3); dizzy spells (No4); and feeling confused (No5). These symptoms in COPD express cognitive issues due to often desaturations and chronic hypoxemia which, in turn, leads to structural and chemical alterations in the central nervous system [38]. The above findings differ from those of van Dixhoorn and Duivenvoorden [13]. Specifically, in the present study, the third factor (central tetany) shares only three items—No3, No4, and No5—with that of the HVS study”. In the HVS validation study two more items, “chest pain” (No1) and “bloated abdominal sensation” (No9) were under the factor of central tetany. In our study, these two items were grouped under the first factor of respiratory symptoms. In COPD, chest pain and bloated abdominal sensation are not considered symptoms of central tetany in COPD, as central tetany is caused by hypoxemia rather than hypocapnia, as seen in HVS.

In the current study, differences were found between groups for all levels of COPD severity, with the highest NQ score for very severe COPD. It is very interesting to mention that DB was detected (NQ score “>23”) in all patients with very severe COPD (12/12), in 30/32 patients with severe COPD, in 7/34 patients with moderate COPD and in 0/6 patients with mild COPD. Pulmonary hyperinflation associated with dyspnea, which triggers DB, is present in patients with stable COPD in percentages of 19.4% in stage I, 68.5% in stage II, 95.3% in stage III, and 100.0% in stage IV [39]. Moreover, patients with severe COPD change their respiratory pattern more than those with moderate COPD, decreasing the inspiration time, and exhibiting rapid and shallow breathing, the main breathing abnormality in COPD (DB) [40,41].

Divergent validity testing revealed a positive correlation between dysfunctional breathing (NQ score) and COPD mortality (BODE index) in the specific COPD sample. This indicates that the higher the NQ score the higher the level of COPD severity, the frequent acute exacerbations, and the risk of hospitalization and death [18,42]. Consequently, dysfunctional breathing (NQ score) may have a direct positive relationship with COPD severity (BODE index). This is a promising finding as the BODE index is expected to replace the GOLD classification for assessing COPD severity in the future [43].

Further, a positive correlation was found between dysfunctional breathing (NQ score) and the impact of COPD on daily life (CAT score). The higher the NQ score, the greater the disease burden, including symptoms such as coughing, phlegm, chest tightness, breathlessness while walking or climbing stairs, and reduced energy and confidence in leaving home. Both the NQ and CAT assess symptoms that affect patients in similar ways and share common underlying mechanisms [44,45]. These findings highlight the importance of addressing dysfunctional breathing in COPD treatment, as the shortness of breath causes a significant impact on daily living [46,47].

Regarding self-perceived dyspnea (Borg scale), the positive correlation with DB (NQ score) supports the interaction between dyspnea and DB [6]. This is in line with the finding of Grammatopoulou et al. for asthma patients [14].

The results of discriminant validity provided a valid cutoff score of >23 for screening DB in patients with stable COPD. Based on that cutoff score, the prevalence DB was found to be 58.3% for the specific sample with stable COPD.

The present study had some limitations: (a) power analysis was not conducted; (b) the sample was convenient; (c) the results of the present study do not provide evidence of external validity for the COPD Greek population and concern only the patients with stable COPD; (d) the diagnosis of dysfunctional breathing was based on the pneumonologists’ diagnosis and not on a “gold standard”; and (e) the construct validity was examined through an exploratory factor analysis with no confirmation of the fit of the model. Future researchers may proceed with a confirmatory factor analysis for further support of our findings.

### Clinical and Research Implementation

The use of a valid tool for screening dysfunctional breathing in COPD will support internal validity in future interventions, help researchers detect patients with dysfunctional breathing, despite the clinicians’ examination [2,27], and support the role of dysfunctional breathing strategies in COPD [4]. Pulmonary rehabilitation uses approaches to reduce dynamic hyperinflation and therefore reduce dysfunctional breathing [48]. Since breathing retraining showed an improvement in asthma symptoms [49,50], it will be of great interest to investigate the effect of such non-pharmaceutic methods on dysfunctional breathing symptoms in COPD with a valid tool.

## 6. Conclusions

The present study offers the first validation evidence for the NQ in Greek outpatients with stable COPD. NQ may be considered a useful tool in clinical and research practice for health professionals to detect dysfunctional breathing in COPD and provide important implications of breathing practices, beyond pharmacology, for the relief of breathlessness that is related to dysfunctional breathing in COPD.

## Figures and Tables

**Table 1 jcm-14-02353-t001:** Item loadings and communalities of the NQ in COPD.

NQ Items	Factor 1 Item Loadings	Factor 2 Item Loadings	Factor 3 Item Loadings	Item Communalities
No1 (Chest pain)	0.96	0.49		0.89
No2 (Feeling tense)	0.96	0.48		0.87
No3 (Blurred vision)			0.54	0.32
No4 (Dizzy spells)			0.72	0.54
No5 (To be confused, losing touch with environment)			0.89	0.85
No6 (Faster or deeper breathing)	0.83			0.74
No7 (Shortness of breath)	0.82			0.71
No8 (Constricted chest)	0.82			0.80
No9 (Bloated feeling in stomach)	0.86			0.76
No10 (Tingling fingers)		−0.91		0.84
No11 (Unable to breathe deeply)	0.90			0.77
No12 (Stiff fingers or arms)		−0.69		0.55
No13 (Tight feelings round mouth)		−0.83		0.78
No14 (Cold hands or feet)		−0.95		0.87
No15 (Palpitations)	0.80			0.73
No16 (Anxiety)	0.99			0.90
Eigenvalue	8.55	1.83	1.57	
% explained variability	53.44	11.43	9.83	
Total explained variability %: 74.70%				

**Table 2 jcm-14-02353-t002:** Differences between participants of different genders, DB diagnosis, COPD levels of severity, follow-up visits, and smoking for the NQ total score.

Variables	*N*	Mean (SD) NQ Score	*F/t*	*p*
Gender				
Male	55	28.27 (13.51)		
Female	29	25.31 (10.84)	*t* = −1.02	0.31
DB diagnosis				
No DB	35	14.69 (6.11)		
DB	49	36.67 (6.65)	*t* = −15.52	<0.001
COPD severity				
Mild	6	6.50 (3.45)		
Moderate	34	18.56 (7.62)		
Severe	32	33.94 (5.53)		
Very severe	12	44.42 (12.66)	*F* = 88.47	<0.001
Follow-up visits				
Every 1–3 months	19	22.42		
In deterioration of symptoms	65	28.66	*t* = −1.92	0.06
Smoking				
No	79	26.57		
Yes	5	38.00	*t* = −1.99	0.05

**Table 3 jcm-14-02353-t003:** Intercorrelations between the NQ total score and BODE index, CAT and Borg.

	NQ	BODE	BORG Scale	CAT
NQ	1.00	−0.81 **	0.47 *	−0.49 *
BODE		1.00	0.62 **	0.62 **
BORG scale			1.00	0.51 **
CAT				1.00

** *p* < 0.001. * *p* < 0.05.

**Table 4 jcm-14-02353-t004:** Screening accuracy of the NQ, based on different cut-off points.

Cut-Offpoints	Sensitivity (%)	Specificity (%)	Positive Likelihoodratio	Negative Likelihoodratio	Positive Predictive Value (%)	Negative Predictive Value (%)
>20	97.96	74.29	4.62	0.027	84.2	96.3
>21	97.96	82.86	6.34	0.025	88.9	96.7
>22	95.92	85.71	7.76	0.048	90.4	93.7
**>23**	**95.92**	**94.29**	**31.06**	**0.043**	**95.9**	**94.3**
>25	91.84	94.29	29.76	0.87	95.7	89.2
>27	87.76	97.14	30.71	0.13	97.7	85.00
>29	79.59	97.14	27.86	0.1	97.5	77.3

## Data Availability

The management of physical and digital data follows European legislation on personal data. They will be kept for two years at the UNIWA and will be destroyed or deleted afterward according to the rules of The Ethics Committee of UNIWA. The datasets presented in this article are not readily available due to time constraints.

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
