# Peer review of "Dysfunctional Breathing, in COPD: A Validation Study"

_jcm, 2025, doi:10.3390/jcm14072353_

Round 1
Reviewer 1 Report
Comments and Suggestions for Authors
I thank the Editor and the Authors to let me review this interesting paper on “Dysfunctional Breathing, in COPD: A Validation Study.”
This manuscript presents a relevant study addressing an important clinical question, with clearly presented results and well-structured tables and figures.
However, several methodological and interpretative issues need to be addressed before the paper can be considered for publication.
1. The sample size justification is not clearly stated, and it is important to clarify whether a power analysis was conducted to determine an appropriate sample size.
2. Confounding factors may not have been adequately accounted for, and if only univariate analyses were used, the study may be affected by selection bias and unmeasured confounders; the authors should consider adjusting for relevant covariates using multivariable regression models or Propensity Score Matching to strengthen causal inferences.
3. Some conclusions appear stronger than what the data supports, and the authors should clarify whether their findings indicate correlation rather than causation, particularly if the study is observational.
4. The study rightly concentrates on the Nijmegen Questionnaire's internal validation for COPD patients, which is a crucial initial step in determining its applicability. The authors should specifically note this as a restriction and talk about the necessity of future validation on external cohorts, since the absence of an independent dataset makes external validation impractical in this study.
5. Clear legends and more context would improve reading as several figures and tables lack comprehensive captions or explanations of data changes.
The manuscript is generally well-written and conveys the study's objectives and findings clearly. However, certain sections, particularly in the methods and results, would benefit from improved clarity and conciseness.
Author Response
Thank you for your fruitful comments, that will help with the improvement of our work.
Having considered your comments, we have made amendments and corrections to the text, which we justify with the following replies:
Comment No1: Yes, we agree with the comment provided by the reviewer. Power analysis was not conducted and may constitute a limitation (added in the limitations of the discussion section). However, in depth screening of the data and careful examination of our literature revealed that the sample size was sufficient. Specifically, Stevens stated that the minimum sample size may range from two subjects per variable to 10 subjects per variable (2002). Monte Carlo study by Guadagnoli ana Velicer (1998) recommended that components with four or more loading above 0.60 in absolute value are reliable, regardless of sample size. Another Monte Carlo study by Mundfrom, Shaw, and Ke (2005) showed that if communalities are high and the number of factors is small, factor analysis can be reliable for sample sizes well below 50. As communalities increase, the influence of sample size declines and has relatively little impact on quality solutions, meaning that accurate recovery of population solutions may be obtained using a fairy small sample (MacCallum, Widaman, Zhang and Hong, 1999). In addition, with loadings of 0.80, the minimum sample size decreases, requiring one third the sample size of 0.50 loadings.
In the present study, we had a sample of 84 people with COPD, most item loadings and communalities were higher than 0.70 and the KMO (Kaiser-Meyer-Olkin) measure for the overall data set was found to be 0.916 confirming the sample adequacy for the analysis.
Comment No2: Well, certainly there are numerous analyses that may be conducted, we strongly agree with the reviewer. However, our theory suggests that regression models are relevant to concurrent validity as such, and not a way to provide construct validity evidence. In any case, let us consider the following:
What exact variable (s) may be used as confound variables and what evidence is there in the literature to use them as such? To our knowledge, severity is the most prominent variable that has an impact upon the DB of COPD patients. In the present study, severity was used as an independent variable to test the construct validity of our data.
We would be grateful if the reviewers suggested exact variables that may be used as a covariate and guide us to the next steps. See, we carefully selected our variables, based on our literature review, and we may not start analyzing them otherwise using covariates that have not been substantiated in the literature so far.
Comment No3: We agree with the reviewer; we proceeded with a shorter discussion focusing on the main results with no causal information.
Comment No4: Well, yes, we agree. To our understanding, the sentence requires a brief addition. For example: The use of a valid tool for screening dysfunctional breathing in COPD will support internal validity in future interventions, looks more appropriate. Our validity and reliability evidence theory suggests that specific evidence for validity and reliability may be provided for each special population and study conducted. Based on that, and following our literature review (selection of the appropriate variables), we consider that our findings have sufficient evidence of internal validity. With respect to the external validity however, the interpretation is different. See, we do not have a wide Greek data set to relate our findings with. We would be very careful to generalize our findings to the COPD Greek population right now. Therefore, we agree with the comment addressing the limitation in the external validity of the present study. We added the limitation in the discussion.
Comment No5: Well, no objection, but we really need to know what to change in the manuscript. There are four tables, following the exact format used in validation studies. In our attempt, we tried to avoid adding more tables (e.g. socio-demographic characteristics, descriptive statistics etc.) and keep the manuscript simple. In any case, we have no objection, but we need to ask you to be more specific. Anyway, we proceeded with a new format with clearer legends in all tables.

Reviewer 2 Report
Comments and Suggestions for Authors
A brief summary: Article about known diagnostic method in specific group of patients (with COPD).
General concept comments: Article is well structured, interesting, clearly described with need for small improvements. It subline important problem in patients with COPD. There was no articles about this method in this group of patients before.
Introduction: Clear introduction, show general view of the problem and indicate aims for the study
Material and methods: Clear and accurate. I advise to add to patients characterisation data about GOLD COPD groups A, B, E.
Results: Demonstration of results need small improvements. Tables are a bit unclear.
Advise to add a table which compare two groups of patients: people with COPD and with or without DB.
Discussion: Clear discussion with scientific literature.
Conclusions are consistent with evidence and arguments presented.
Citations are mostly older than 5 years but are relevant. There is no excessive number of self-citations
Author Response
Thank you for your fruitful comments, that will help with the improvement of our work.
Having considered your comments, we have made amendments and corrections to the text, which we justify with the following replies:
Comment No1: Material and methods: Well, in table 2, we presented the descriptives (Ms, SDs, p) of the NQ score for the participants of all COPD severity levels. The differences between these groups were described in detail. We have difficulty, we could not understand so well what the meaning of your specific comment was. Could you please repeat or attempt to clarify?
Comment No1: Results: To our understanding, the reviewer wants to add a table presenting the differences between COPD patients either with or without DB, with respect to their NQ scores. In table 2, this information is already incorporated into the manuscript, in the results and discussion sections. Please clarify.

Reviewer 3 Report
Comments and Suggestions for Authors
The authors investigated the validity of the Nijmegen Questionnaire for dysfunctional breathing in a cohort of 84 patients with stable COPD in Greece. They assessed different dimensions of validit and provided evidence for the overall validity and reliability of the NQ in individuals with stable COPD. The study is methodologically sound and its results may help to further improve COPD patients' management. However, I have some comments regarding the presentation of the findings.
- How did the authors define the inclusion criterion "stable COPD"?
- The description of the cohort in the Methods section (lines 62-68) should be moved to the Results section.
- The report of the mean scores of the population (section 4.1) could be better displayed in a table.
- Tables 2 and 3 require re-formatting.
- Section 4.2, line 181-182: "NQ total score reached statistical significance (p = 0.05) between smokers and non-181 smokers with COPD." - Generally, the significance level is set to be < 0.05. Thus, p=0.05 is not statistically significant. This needs to either be corrected (including the resulting conclusions), or explained why a non-standard significance level has been used.
- Section 4.2, Discriminant validity testing: I suggest to additionally present the ROC curve, potentially in the appendix.
- Section 4.2, line 196: The numbers in the text to not match the numbers in the table and need to be corrected.
- Section 4.2, line 197: I assume that "AUC = 0.05" must be replaced with "AUC = 0.5"?
- Discussion: The discussion is rather lengthy and could be shortened to discuss the main results.
- The sample of 84 patients is rather small for a validity study, which should be mentioned in the limitations. Also: What do the authors mean by "The sample was convenient."?
- Abstract: The phrase "To validate the NQ in individuals with COPD." should be rephrased in a full sentence.
- Introduction, line 36: I assume "time-Ti" is to be "time"?
Author Response
Thank you for your fruitful comments, that will help with the improvement of our work.
Having considered your comments, we have made amendments and corrections to the text, which we justify with the following replies:
Comment No1: This inclusion criterion has the citation No16 in the manuscript. Patients with stable COPD are those not experiencing acute exacerbation (added in the method section, 2.1. study population).
Comment No2: Well, as we see in sever validation papers, the description of the cohort is usually presented in the method section, as it incorporates its characteristics recorded in the recruitment. In the results section, we displayed the results of the measured variables. In the case of an RTC, it is mandatory to present the characteristics of the sample in a table, to support the homogeneity of the sample, but not in a validation study. If the reviewer still insists on his comment, we will have no objection.
Comment No3: Please allow us to mention that there are already four tables, following the exact format used in validation studies. In our attempt, we tried to avoid adding more tables (e.g. socio-demographic characteristics, descriptive statistics etc.) and keep the manuscript simple. If the reviewer still insists on his comment, we will have no objection.
Comment No4: We agree with your comment; all tables are re-formed.
Comment No5: We agree, and we proceeded with the correction, as mentioned.
Comment No6: We agree, and we will submit the figure of the ROC curve; it should be added in the appendix.
Comment No7: Thank you for your comment; the numbers are checked and corrected in the text.
Comment No8: There is no doubt of AUC = 0.5 and we corrected it at once. Thank you for the comment.
Comment No9: We agree, and we proceeded with a shorter discussion, focusing on the main results.
Comment No10: We agree with the reviewer, and we deleted the word “adequate. We used that word, because the sample size was based on the rule of thumb of minimum five individuals per item. Stevens stated that the minimum sample size may range from two subjects per variable to 10 subjects per variable (2002). Monte Carlo study by Guadagnoli ana Velicer (1998) recommended that components with four or more loading above 0.60 in absolute value are reliable, regardless of sample size. Another Monte Carlo study by Mundfrom, Shaw, and Ke (2005) showed that if communalities are high and the number of factors is small, factor analysis can be reliable for sample sizes well below 50.
Beyond the limitation for the power analysis (added in the discussion section), in the present study, the KMO (Kaiser-Meyer-Olkin) measure for the overall data set was found to be 0.916 confirming the sample adequacy for the analysis.
Comments on the Quality of English Language
Comment No1: Abstract: Well, we agree with your comment, and we rephrased it in a full sentence, but please allow me to say it was due to the formatting of the manuscript in publication draft by the journal services.
Comment No2: Introduction: Ti is the abbreviation of the inspiratory time, but there is no objection, since it does not appear elsewhere in the text; we omitted it.

Round 2
Reviewer 3 Report
Comments and Suggestions for Authors
I congratulate the authors to their revision and have no further concerns.